# Assessment of carbon monoxide in exhaled breath using smokerlyzer among in-school adolescents in Osun State: A cross sectional study

**Funmito Omolola Fehintola**[1]*, Adesola Olumide[2], Adesegun Fatusi[1], Olayemi Omotade[2]

**1** Department of Community Health Faculty of Clinical Sciences Obafemi Awolowo University Ile-Ife, Nigeria, **2** Institute of Child Health Faculty of Public Health, College of Medicine University of Ibadan, Nigeria

* funmitoabioye@yahoo.com

## Abstract

### Background

Smoking is one of the modifiable risk factors of non-communicable diseases and cause of premature death. Surveillance is most often based on self-reported data among adolescents, but few studies have measured exhale breath using Carbon Monoxide in them. This study aimed to measure the prevalence of smoking by measuring exhaled Breath Carbon Monoxide Threshold Levels and Self-Reported Cigarette Smoking in a sample of in-school adolescents in Osun State Nigeria.

### Methods

The study employed a descriptive, cross sectional study design, a multi-stage sampling technique was used to select 616 adolescents from selected secondary schools in Osun State. Smoking status of adolescents were measured using Smokerlyzer, and WHO STEPwise questionnaire. The Cut-off point for current smokers using the exhale level of carbon monoxide (eBco) was ≥ 10parts per million. The collected data were entered into SPSS version 20 software for statistical analysis. Multivariable logistic regression analyses were done to identify the true effect of predictor variables on the outcome variable. Statistical significance was declared at a p-value < 0.05.

### Findings

Smoking by self-report was 80 (13.0%) while the smokerlyzer detected 168 (27.3%) current smokers. Cohen's κ = 0.063 (smokerlyzer) showed poor agreement between self-report and objective measures. Determinants of smoking include older age (AOR = 1.59, CI = 1.01–2.34), class level, compared with students in JSS1, SSS1 (AOR = 0.14, 95% CI: 0.07–0.26), and SSS2 (AOR = 0.46, 95% CI: 0.23–0.78) were

**Data availability statement:** Data cannot be made publicly available due to ethical considerations as per regulations imposed by the Consortium for Advance Research Training in Africa. Requests for data can be dispatched to the Consortium for Advance Research Training in Africa via email at carta@aphrc.org. Data is also available from this paper's Corresponding Author.

**Funding:** This research was funded by the Consortium for Advanced Research Training in Africa (CARTA). CARTA is jointly led by the African Population and Health Research Center and the University of the Witwatersrand. CARTA is funded by: The Carnegie Corporation of New York [grant no. G-19-57145], The Sida [grant no. 54100113], and by the Uppsala Monitoring Centre and the DELTAS Africa Initiative [grant no. 107768/Z/15/Z]. The DELTAS Africa Initiative is an independent funding scheme of the African Academy of Sciences (AAS)'s Alliance for Accelerating Excellence in Science in Africa (AESA) and supported by the New Partnership for Africa's Development Planning and Coordinating Agency (NEPAD Agency) with funding from the Wellcome Trust (UK) and the UK government. The statements made and views expressed are solely the responsibility of the authors. The funders had no role in study design, data collection and analysis, decision to publish, or preparation of the manuscript.

**Competing interests:** The authors have declared that no competing interest exist.

significantly less likely to smoke. Rank regression analysis revealed that age was negatively associated with the rank of smoking status (B = −0.10, p = 0.008).

## Conclusion

Prevalence of smoking using smokerlyzer was higher compared to self-report among in-school adolescents. The study identified older age and class as important predictors of smoking Targeted health education should be done in schools to enhance smoking cessation.

## Introduction

Smoking is a major public health concern, resulting into to numerous health problems and complications worldwide. Globally, approximately 1.3 billion people use tobacco, making it one of the leading causes of preventable diseases [1]. Around 80% of the 1.3 billion tobacco users worldwide live in low- and middle-income countries, where the burden of tobacco-related illness and death is heaviest [2].The economic costs of tobacco use are substantial and include significant health care costs for treating the diseases caused by tobacco use as well as the lost human capital that results from tobacco-attributable morbidity and mortality [1].

Nigeria, the most populous country in Africa, has 22% of its population to be adolescents and has one of the leading tobacco markets in Africa, with over 18 billion cigarettes sold annually [3]. Adolescents in Nigeria are particularly vulnerable. Every year, more than 16,100 of Nigeria's population die from tobacco-caused disease; 748,800 Nigerians aged 15 years or older and more than 25,000 aged 10–14 years use tobacco every day [4]. Nigeria signed the WHO FCTC in 2004, ratified it in 2005, and domesticated it through the National Tobacco Control Act of 2015 [5,6].

Adolescents who smoke are at an increased risk of developing long-term health problems and are more likely to continue smoking into adulthood, further exacerbating the burden of tobacco-related diseases [7–9]. Active smoking causes reduced lung function and impaired lung growth during childhood and adolescence, and the early onset of lung function decline during late adolescence and early adulthood [10]. They are prone to cardiovascular diseases, asthma, addiction to nicotine and dental health issues [11]. Furthermore, adolescents that smokes are at risk of behavioural issues such as risky sexual behaviours, poor dietary habits and alcohol abuse [12,13]. Early abdominal aortic atherosclerosis, which affects the flow of blood to vital organs, has been found among young smokers [14]. This leads to consequences such as hypertension, ischemic heart disease, and chronic obstructive pulmonary disease later in life [15].

Several studies have been done among adolescents on smoking using self –report [16–18]. However, concerns have been raised about measuring smoking by self-report among adolescents [19,20]. Adolescents are usually fearful of adults' judgment and disapproval, which may prevent them from revealing their smoking status [18,21]. Also, adolescents report ever smoking rather than admit recent smoking

[22,23]. This is commonly seen in school settings where smoking is not allowed. Hence it is possible that self-report will underestimate smoking [23,24].

Portable tools like the smokerlyzer that quantify the amount of carbon monoxide exhaled (eBCO) can help objectively measure the current status of smoking among adolescents [25,26]. This will help to identify the true prevalence of smoking and help design targeted intervention for these age group. This study aims to determine the prevalence of smokers among in-school adolescents using smokerlyzer. It also helps to identify the predictors of smoking among adolescents.

## Methods

The State of Osun in Nigeria served as the study's location. The state has thirty Local Government Areas (LGAs), as well as the Ife-Modakeke area office is present, and Osogbo is the state's capital. According to the 2006 Census, roughly 3,423,535 people live in the state [27]. The estimated population for 2021 is 5,926,721, based on the 2.8% annual growth rate. The Yoruba comprise most of the state's population [27]. The study employed cross-sectional study design. The study population were in-school adolescents. Adolescents aged 10–19 years who were in in JSS2 (grade 8) to SSS3 (grade 12) were included in the study. We excluded adolescents' ages 10–19 years who are yet to spend up to a session in school. Adolescents whose parents do not give consent and adolescents who did not provide assent Adolescents who use firewood, or have pulmonary lung diseases, and such were excluded from participating in the objective assessment

### Sample size determination

This study was part of a multiphase study where the initial sample size was determined using the formula $(Z\alpha + Z\beta)^2 P (1 - P) d^2$ [28]. P = estimate of true proportion in the population; was taken as the prevalence of media health literacy among secondary school students in Osun State 37.7% [29]. Giving a design factor of 1.5 due to clustering and non-response rate (NRR) of 10% we arrived at a sample size of 1232 [30]. A subset of the initial sample size (50%) was used for the objective measurement of smoking due to the financial constraints. A predetermined number of students were allotted to each selected school and respondents were selected using simple random sampling technique.

### Sampling technique for the survey

All secondary public and private schools in the Osun East Senatorial District were included in the sampling frame. The study locations and respondents were chosen using a multistage sampling procedure. In Stage 1: One senatorial district (Osun East Senatorial district) was randomly selected by a simple random sampling technique by balloting from the three senatorial districts in Osun State. Stage 2: The balloting system randomly chose two rural LGAs. Due to more schools in urban than rural areas, the number of rural LGAs selected was twice that of urban areas. Stage 3: we chose five public secondary schools and five private secondary schools from each LGA. Stage 4: An arm of the class was chosen using a simple random sample procedure from each stratum of classes JSSS 2 to SSS3 (corresponding to Grades 8–12), making five classes from each school. For some private schools with only one arm, the available were selected. Stage 5: Based on the predefined number of participants allotted to the school, a proportionate sample was drawn from the chosen courses.

### Study instrument

WHO STEPwise questionnaire for chronic disease risk factor surveillance (STEPS). Using this tool, we assessed adolescents' lifestyles regarding smoking

### Objective assessment of smoking habits of adolescents

**A piCO+ smokerlyzer®.** A piCO+ smokerlyzer® is a simple, easily movable, cheap instrument for measuring the eBco (exhale carbon monoxide). It was used to assess the smoking status of respondents. It is manufactured in England,

United Kingdom, and measures the exhaled breath of carbon monoxide in parts per million. In this investigation, the manufacturer-recommended standard smoking threshold points for adolescent and adult age groups were utilised to calibrate the piCO+ smokerlyzer® to detect eBCO levels. The recommended cut-off numbers for teenagers who smoke are:zero to four ppm for non-smokers, five to six ppm for kids who are smoking light or sometimes, seven to ten ppm for smokers,eleven to fifteen ppm for frequent smokers, and sixteen to twenty-five ppm for addicted smokers.

**Data collection procedures.** Data collection took place between July 1 and December 15, 2022. Data was collected using self-administered questionnaire. All the administered questionnaires were collected and checked daily for errors and completeness, and appropriate corrections were made

**Procedure for assessment of smoking.** Fifty percent of the survey sample was selected for the objective assessment. Students were first assessed for their cigarette smoking habits using the self-administered questionnaire. The eBCO test was then attended by a trained research assistant and recorded on their questionnaire. All eBCO tests were conducted in the school hall between 10 am and 12noon. Testing was conducted in line with the manufacturer's guidelines. Before each student was tested, a clean cardboard mouthpiece and a breath sampling D-piece were fitted to the piCO+ smokerlyzer®.

Students were asked to inhale and hold their breath whilst a 15-s countdown displayed. In order to empty lungs completely, at the end of the 15 s, students were coached to blow through the mouthpiece slowly and fully, as soon as the machine started beeping. Adolescents were told the test results immediately at the cessation of the test. The adolescents were categorized as smokers, non-smokers, frequent smokers, and addicted smokers based on the findings from the test. To minimize the drawback of environmental exposure, selected students were asked whether they use firewood, or have pulmonary lung diseases, and such were excluded from participating in the objective assessment

**Data management and analysis plan.** After carefully correcting the data, they were exported to SPSS software, and analysis was performed using SPSS version 20.

**Univariate analysis:** For categorical independent variables, including sex, age group, religion, ethnicity, and categorical school factors (such as school ownership), frequencies and proportions were reported. Means, standard deviations, and medians were used to summarize quantitative independent variables such as gender in years. We calculated the prevalence of smoking.

**Bivariate analysis:** Using Pearson's chi-squared test, this examined the connection between smoking and sociodemographic factors. This statistical test also determined the socio-demographic and educational influences on smoking. The statistical significance level was set at p-values less than 0.05.

**Binary logistic regression:** Because the outcome variable is smoking, binary, binary logistic regression was applied. We determined the association between smoking and individual-level factors, including socio-economic status, religion, ethnicity, gender, and age group.

**Rank regression**: In addition, a rank regression analysis was done, treating the outcome variable as a continuous score for a robust analysis.

**Measures and outcome variables.**

**Self-reported smoking**: The students were asked if they had ever smoked. They were also asked if they had smoked in the last 30 days.

**Smoking (objective assessment)**: This was assessed using the amount of exhaled carbon monoxide in parts per million

## Ethical considerations

The ethical review board of the Institute of Public Health (IPH) at Obafemi Awolowo University in Ile-Ife granted ethical approval. HREC number is NHREC/IPH/OAU12/1951.This study was conducted in accordance with the ethical standards outlined in the

Declaration of Helsinki and its subsequent amendments or comparable ethical standards. The Osun State Ministry of Education permitted the study to be conducted. The authorities of the chosen schools also gave their approval. Informed written consent was obtained from parents and guardians of respondents less than 18 years old. Assent was also obtained from respondents less than 18years old. Informed written consent was obtained from respondents 18 years and above.

### Assessment of safety

The study did not pose any health risks to the participants.

## Results

### The socio-demographic characteristics of the respondents

Three hundred and seventy-six (61.0%) respondents were aged 15–19. The average age of respondents was 15.3±2.0 years. Half of the respondents, 315 (51.1%), are males, while 415 (67.4%) of the respondents are from public schools. Two hundred and one (32.6%) respondents are from private schools. Three hundred and eighteen (51.6%) of respondents are from rural local governments (Table 1).

### Smoking habits among respondents' self-report

Eighty (13%) of in-school adolescents said they are currently smoking. The mean age at smoking is 12.4±2.7 years. Fifty-nine (73.8%) of the respondents who are current smokers said they had friends who smoke. (Table 2)

### Categorization of smokers by smokerlyzer

Table 3 shows that out the 168 respondents identified as smokers by smokerlyzer 7(4.2%) are light smokers, 124(73.8%) are smokers, 22(13.1%) are frequent smokers and 15(8.9%) are addicted smokers.

### Smoking status of respondents by school, sex and location

Table 4 shows that by self-report, 50 (12.0%) of the respondents from public schools were smokers, and 30 (14.9%) from private schools were smokers. By smokerlyzer, 100 (24.1%) of the respondents from public schools were smokers, while 68 (33.8%) of the respondents from private schools were smokers. A statistically significant association exists between smoking (smokerlyzer) and school type ($p < 0.001$). By self-report, fifty-seven (17.9%) of the respondents from secondary schools in rural local government areas were smokers, while 23 (7.7%) from secondary schools in Urban were smokers. A statistically significant relationship exists between smoking and school location ($p < 0.001$). By self-report, fifty-five (17.5%) of the respondents that were males were smokers, while 25 (8.3%) of respondents that were females were smokers. By smokerlyzer, 124 (39.4%) of respondents that were males were smokers, while 44 (14.6%) of respondents that were smokers were female. There is a statistically significant association between smoking and sex ($p < 0.001$).

### Smoking status by self-report and smokerlyzer among selected in-school adolescents'

Table 5 shows that there was no significant agreement between smoking by self-report and by smokerlyzer (Cohen's=0.048) indicating slight agreement between the two measurement methods. This level of agreement was not statistically significant, suggesting that the classification of smoking status differed between self-report and biochemical verification in this sample.

### Factors associated with smoking among in-school adolescents in Osun State

Table 6 shows that one-hundred and twenty-four of the respondents were male smokers, while 44 (14.6%) of respondents were females were smokers. There is a statistically significant relationship between sex and respondents' smoking status

**Table 1. Sociodemographic characteristics of respondents.**

| Variables | Frequency(N=616) | Percentages |
|---|---|---|
| **Age** | | |
| Mean Age:15.3±2years | | |
| 10–14 years | 240 | 38.9 |
| 15–19 years | 376 | 61.0 |
| **Sex** | | |
| Male | 315 | 51.1 |
| Female | 301 | 48.9 |
| **School type** | | |
| Public | 415 | 67.4 |
| Private | 201 | 32.6 |
| **Ethnic group** | | |
| Yoruba | 554 | 89.9 |
| Non-Yoruba | 62 | 10.1 |
| **Religion** | | |
| Christian | 486 | 78.9 |
| Muslim | 87 | 14.1 |
| Tradition | 43 | 7.0 |
| **Class** | | |
| JSS2 | 53 | 8.6 |
| JSS3 | 98 | 15.9 |
| SSS1 | 162 | 26.3 |
| SSS2 | 138 | 22.4 |
| SS3 | 165 | 26.8 |
| **Location** | | |
| Urban | 298 | 48.4 |
| Rural | 318 | 51.6 |

(p<0.001). One hundred and twenty-six (30.4%) of respondents in public schools and 42 (20.9%) of respondents from private schools were smokers. There is a statistically significant relationship between the smoking status of respondents and the type of school (p=0.013). There is a statistically significant relationship between school location and smoking (p<0.001).

### Binary logistic regression modelling outcomes with factors associated with smoking among in-school adolescents

After adjusting for all covariates in the model, age was significantly associated with smoking. Adolescents aged 15–19 years had 1.59 times higher odds of being current smokers compared with those aged 10–14 years (AOR=1.59, 95% CI: 1.01–2.34, p=0.020).

Class level also showed a strong and consistent protective association with smoking. Compared with students in JSS1, those in JSS2 (AOR=0.39, 95% CI: 0.19–0.84, p=0.015), JSS3 (AOR=0.41, 95% CI: 0.23–0.72, p=0.002), SSS1 (AOR=0.14, 95% CI: 0.07–0.26, p<0.001), and SSS2 (AOR=0.46, 95% CI: 0.23–0.78, p=0.004) were significantly less likely to smoke. The magnitude of this protective effect was greatest among students in SSS1, whose odds of smoking were reduced by approximately 86% relative to JSS1 students.

**Table 2. Adolescents' smoking habits (self-report).**

| Variables | Frequency (n = 616) | Percentages (%) |
|---|---|---|
| **Adolescent currently smoking** | | |
| Yes | 80 | 13.0 |
| No | 536 | 87.0 |
| **Age at first cigarette smoking(n = 80)** | | |
| Mean smoking age = 12.4 ± 2.7 years | | |
| 10–14years | 42 | 52.5 |
| 15–19 years | 38 | 47.5 |
| **In the past month, number of days adolescent used any tobacco products** | | |
| 1–7days | 52 | 65.0 |
| 8–14days | 23 | 28.8 |
| 15–21days | 4 | 5.0 |
| 22–30days | 1 | 1.2 |
| **Adolescent had friends who smoked:** | | |
| Yes | 59 | 73.8 |
| No | 21 | 26.2 |
| **School regulation Against smoking** | | |
| Yes | 395 | 64.1 |
| No | 221 | 35.9 |
| **School has handbill against Smoking** | | |
| Yes | 228 | 37.0 |
| No | 388 | 63.0 |
| **School health Clubs** | | |
| Yes | 200 | 32.5 |
| No | 416 | 67.5 |

**Table 3. Smokerlyzer test result among selected in-school adolescents'.**

| Variable | Frequency | percentages |
|---|---|---|
| Light/ Casual smokers | 7 | 4.2% |
| Smoker | 124 | 73.8% |
| Frequent smoker | 22 | 13.1% |
| Addicted smoker | 15 | 8.9% |

Sex, school type, location, presence of school rules against smoking, presence of anti-smoking billboards/posters, and the existence of punishments for smoking were not significantly associated with smoking in the adjusted model (all p > 0.05) (Table 7).

## Rank regression

The rank regression model examining predictors of the relative ordering of smoking status among in-school adolescents in Osun State is presented in Table 8. Age was negatively associated with the rank of smoking status (B = −0.10, p = 0.008),

**Table 4. Description of smoking status of respondents by school, sex and location (N = 616).**

| Variable | Self-report | | N | χ²; p-value | Smokerlyzer | | N | χ²; p-value |
|---|---|---|---|---|---|---|---|---|
| | Smoking n (%) | | | Not smoking n (%) | Smoking n (%) | Not smoking n (%) | | |
| **School type:** | | | | | | | | |
| Public | 50(12.0) | 365(88.0) | 415 | | 100(24.1) | 315(75.9) | 415 | |
| Private | 30(14.9) | 171(85.1) | 201 | 284.11; <0.001 | 68(33.8) | 133(66.2) | 201 | 159.29; <0.001 |
| **Location:** | | | | | | | | |
| Rural | 57(17.9) | 261(82.1) | 318 | | 93(29.3) | 225(70.7) | 318 | |
| Urban | 23(7.7) | 275(92.3) | 298 | 199.45; <0.001 | 75(25.2) | 223(74.8) | 298 | 75.0; <0.001 |
| **Sex:** | | | | | | | | |
| Male | 55(17.5) | 260(82.5) | 315 | | 124(39.4) | 191(60.6) | 315 | |
| Female | 25(8.3) | 276(91.7) | 301 | 193.77; <0.001 | 44(14.6) | 257(85.4) | 301 | 91.95; <0.001 |

χ²= McNemer Chisquare.

**Table 5. Comparison of smoking status by self-report and smokerlyzer among selected in-school adolescents' (N=616).**

| Smokerlyzer | Self-report | | | | |
|---|---|---|---|---|---|
| | Smoking | Not smoking | N | Kappa; | p-value |
| | n (%) | n (%) | | | |
| **Smoking** | 50 | 118 | 168 | 0.048 | 0.063 |
| **Not smoking** | 30 | 418 | 448 | | |
| **Total** | 80 | 536 | 616 | | |

indicating that increasing age was associated with a lower relative position in the distribution of smoking status. Sex rank also showed a negative association (B= −0.078, p=0.041), suggesting differences in smoking rank across sex categories based on the ranking scheme used.

In contrast, class rank demonstrated a positive association with smoking rank (B=0.145, p<0.0001), suggesting that adolescents in higher classes tended to occupy higher relative positions in the ranked smoking outcome. School type, location, presence of school rules against smoking, availability of anti-smoking billboards or posters, and school punishment for smoking were not significantly associated with the ranked smoking outcome.

## Discussion

According to self-report, the prevalence of smoking in this study was 13.0%. It is similar to the findings of a descriptive study conducted among students in rural and urban secondary schools in Enugu State, Nigeria, where the current smoking prevalence was 13.3% [31]. It was also similar to the findings of a study conducted among secondary school students in Pakistan, which estimated the current smoking prevalence to be 13.7% [32]. In contrast, findings from the in-school adolescents in Osun state were lower than a prevalence of 21.1% reported for current smoking among adolescents in Eastern Ethiopia [33]. The differences in the current level of smoking might be due to variation in geographical location and methodological design of the studies. Other studies conducted in the Southwestern geopolitical zone in Nigeria have also reported relatively lower rates of current cigarette smoking compared to counterparts in India, for instance [33].

More males engaged in cigarette smoking than their female counterpart. This result is the same as that of a study of middle and high school students in Chongqing Province, country where tobacco use by boys was significantly higher than by girls [34]. Similar reports were obtained from a multi-country study on prevalence of smoking by age and sex between

**Table 6. Factors associated with smoking (objectively assessed) among in-school adolescents in Osun State.**

| Variables | Smoking(168) | Non-smoking(448) | χ²;p-value |
|---|---|---|---|
| **Age group** | | | |
| 10–14 years | 71(29.6) | 169(70.4) | 1.057 |
| 15–19 years | 97(25.8) | 279(74.2) | 0.304 |
| Sex | | | |
| Male | 124(39.4) | 191(60.6) | 47.525 |
| Female | 44(14.6) | 257(85.4) | <0.001 |
| **School** | | | |
| Public | 126(30.4) | 289(69.6) | 6.107 |
| Private | 42(20.9) | 159(79.1) | 0.013 |
| **Class** | | | |
| JSS2 | 5(10.4) | 48(89.6) | 39.300 |
| JSS3 | 12(12.2) | 86(87.8) | <0.001 |
| SSS1 | 36(22.2) | 126(77.8) | |
| SSS2 | 51(36.9) | 87(63.1) | |
| SSS3 | 64(38.8) | 101(61.2) | |
| **Location** | | | |
| Urban | 61(20.5) | 237(79.5) | 13.469 |
| Rural | 107(33.7) | 211(66.3) | <0.001 |
| **School has rules against smoking** | | | |
| **Yes** | 69 (17.8) | 326 (82.2) | 53.271 |
| **No** | 99 (44.8) | 122 (55.2) | <0.001 |
| **School has Billboards for smoking** | | | |
| **Yes** | 78 (34.2) | 150 (65.8) | 8.238 |
| **No** | 90 (23.2) | 298 (76.8) | 0.004 |
| **School has school Health clubs** | | | |
| **Yes** | 38 (19.0) | 162 (81.0) | 9.610 |
| **No** | 130 (31.3) | 286 (68.7) | 0.002 |

1980 and 2012. Implementing health education interventions and other proven interventions among the study population could result in a reduction in the smoking prevalence.

Our study revealed that almost three-quarters (73.8%) of adolescents that smoke have friends that also smoke. This finding was similar to that obtained by a study among secondary school students in China and Californian [35] which reveals that secondary school students were more likely to be smokers if their peers were smokers. At this age, adolescents tend to believe that their peers are always right and will readily accept whatever they introduce to them [34]. Cigarette smoking is one of the most common behaviours fostered by peer pressure, particularly in low and middle income nations when prohibitive laws against the sale of cigarettes to minors and adolescents are not enforced.

Our findings showed that the smokerlyzers detected more current smokers among the adolescents compared to self-report. Previous studies have also supported this finding that prevalence of current smokers among adolescents detected by smokerlyzer is higher than prevalence by self-report [36,37]. A possible explanation for the under-estimation documented by self-report may be because smoking is not culturally acceptable in Nigeria and hence adolescent report what they think is acceptable to the researcher.

**Table 7. Binary logistic regression modelling outcomes with factors associated with smoking among in-school adolescents.**

| Variables | Smoking status | | p-value |
|---|---|---|---|
| | AOR | (95% C.I OR) | |
| **Age** | | | |
| 10–14 years (Ref) | – | – | – |
| 15–19 years | 1.59 | (1.01–2.34) | 0.02 |
| **Sex** | | | |
| Female (Ref) | – | – | – |
| Male | 1.17 | (0.77–1.78) | 0.469 |
| **School type** | | | |
| Public (Ref) | – | – | – |
| Private | 1.23 | (0.84–1.80) | 0.297 |
| **Class** | | | |
| JSS2 (Ref) | – | – | – |
| JSS3 | 0.39 | (0.19–0.84) | 0.015 |
| SSS1 | 0.41 | (0.23–0.72) | 0.0002 |
| SSS2 | 0.14 | (0.07–0.26) | <0.0001 |
| SSS3 | 0.46 | (0.23–0.78) | 0.004 |
| **Location** | | | |
| Rural (Ref) | – | – | – |
| Urban | 0.86 | (0.55–1.35) | 0.513 |
| **School has rules against smoking** | | | |
| Yes (Ref) | – | – | – |
| No | 0.75 | (0.44–1.28) | 0.291 |
| **School has billboards, posters & handbills preventing smoking** | | | |
| Yes (Ref) | – | – | – |
| No | 0.93 | (0.62–1.39) | 0.727 |
| **School has punishment against smoking:** | | | |
| No (Ref) | – | – | **–** |
| Yes | 0.99 | (0.57–1.72) | 0.979 |

Dependent variable: Smoking status – Non-smoker (0), Smoker (1), reference -Ref.

In this current study there was a relationship between, older age and being involved in smoking habit. This was similar to that of previous studies which revealed that older age is associated with smoking [38–41]. Students that are in the middle or late stage of adolescence may have acquired more independent than the ones in early adolescence, hence their increasing access to cigarette [39].

Although having schools with health clubs, billboards for health information, and laws against smoking, alcohol drinking and eating junks are not significantly associated with smoking in this study. It does in previous studies [29,31]. A study revealed that countries that have adopted A "Programmes" that bans the sale of cigarette in schools reported lower likelihood of engaging in smoking among their adolescents [42]. Also, a study conducted among high school students in Seychelles noted that the establishment of school policies against risky smoking significantly reduced the likelihood of the adolescents engaging in the practice of smoking [31]. This serves to emphasize the value of a safe learning environment in limiting dangerous behaviour in secondary school children.

**Table 8. Rank Regression of relative Smoking status among In-School Adolescents in Osun State.**

| | Unstandardized Coefficients | P-value | 95.0% CI for B | |
| --- | --- | --- | --- | --- |
| | B | | Lower Bound | Upper Bound |
| (Constant) | 307.337 | **<0.0001** | 246.884 | 367.791 |
| Rank of age | -0.1 | **0.008** | -0.174 | -0.026 |
| Rank of School type | -0.029 | 0.423 | -0.1 | 0.042 |
| Rank of Location | 0.031 | 0.45 | -0.049 | 0.11 |
| Rank of Sex | -0.078 | **0.041** | -0.153 | -0.003 |
| Rank of Class | 0.145 | **<0.0001** | 0.081 | 0.209 |
| Rank of School has rules against smoking | 0.079 | 0.119 | -0.02 | 0.179 |
| Rank of School has billboards, posters & handbills preventing smoking | -0.003 | 0.93 | -0.078 | 0.072 |
| Rank of School has punishment against smoking | -0.041 | 0.408 | -0.138 | 0.056 |

Dependent Variable: Rank of Smoking status.

## Conclusion

The findings of our study are promising and suggest that eBCO levels using the piCO+ smokerlyzer® can be an accurate method to identify smokers from non-smokers, especially in the school setting. Finally, the eBCO method could be a more accurate tool when assessing recent tobacco exposure as compared to self-report.

## Strengths and limitations

This study assessed smoking using self-report and objective measure, previous studies in this environment have used mainly self-report. This can help reveal the true prevalence of smoking among in-school adolescents in Osun State. There may be social desirability bias in which respondents give the socially acceptable response to the researcher. To minimize this confidentiality was ensured. These study identified association between individual demographic and academic progression factors and smoking behaviour, however school-level, social economic status, teachers influence and peer pressure were not assessed and could be a potential residual confounders. We suggest that future studies should consider these factors.

## Acknowledgments

The authors thank the participants for their cooperation and support.

## Author contributions

**Conceptualization:** Funmito Omolola Fehintola, Adesola Olumide, Adesegun Fatusi, Olayemi Omotade.

**Data curation:** Funmito Omolola Fehintola, Adesola Olumide, Adesegun Fatusi, Olayemi Omotade.

**Formal analysis:** Funmito Omolola Fehintola.

**Investigation:** Funmito Omolola Fehintola, Adesegun Fatusi.

**Methodology:** Funmito Omolola Fehintola, Adesola Olumide, Adesegun Fatusi.

**Supervision:** Adesola Olumide.

**Validation:** Olayemi Omotade.

**Writing – original draft:** Funmito Omolola Fehintola, Adesola Olumide, Adesegun Fatusi, Olayemi Omotade.

**Writing – review & editing:** Funmito Omolola Fehintola, Adesola Olumide, Adesegun Fatusi, Olayemi Omotade.

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
