## [Decision Letter · Decision Letter 0]

29 Jul 2025

Dear Dr. Fehintola,

Thank you for submitting your manuscript to PLOS ONE. After careful consideration, we feel that it has merit but does not fully meet PLOS ONE’s publication criteria as it currently stands. Therefore, we invite you to submit a revised version of the manuscript that addresses the points raised during the review process.

We look forward to receiving your revised manuscript.

Kind regards,

Billy Morara Tsima, MD MSc

Academic Editor

PLOS ONE

Journal Requirements:

2. In the online submission form you indicate that your data is not available for proprietary reasons and have provided a contact point for accessing this data. Please note that your current contact point is a co-author on this manuscript. According to our Data Policy, the contact point must not be an author on the manuscript and must be an institutional contact, ideally not an individual. Please revise your data statement to a non-author institutional point of contact, such as a data access or ethics committee, and send this to us via return email. Please also include contact information for the third party organization, and please include the full citation of where the data can be found.

3.Thank you for stating the following financial disclosure:  [This research was funded by the Consortium for Advanced Research Training in Africa (CARTA). CARTA is jointly led by the African Population and Health Research Center and the University of the Witwatersrand. CARTA is funded by: The Carnegie Corporation of New York [grant no. G-19-57145], The Sida [grant no. 54100113], and by the Uppsala Monitoring Centre and the DELTAS Africa Initiative [grant no. 107768/Z/15/Z]. The DELTAS Africa Initiative is an independent funding scheme of the African Academy of Sciences (AAS)’s Alliance for Accelerating Excellence in Science in Africa (AESA) and supported by the New Partnership for Africa’s Development Planning and Coordinating Agency (NEPAD Agency) with funding from the Wellcome Trust (UK) and the UK government. The statements made and views expressed are solely the responsibility of the authors.].

Please include this amended Role of Funder statement in your cover letter; we will change the online submission form on your behalf.\

Reviewers' comments:

Reviewer's Responses to Questions

**Comments to the Author**

1. Is the manuscript technically sound, and do the data support the conclusions?

Reviewer #1: Yes

Reviewer #2: Yes

2. Has the statistical analysis been performed appropriately and rigorously?

Reviewer #1: Yes

Reviewer #2: No

3. Have the authors made all data underlying the findings in their manuscript fully available?

Reviewer #1: Yes

Reviewer #2: Yes

4. Is the manuscript presented in an intelligible fashion and written in standard English?

Reviewer #1: Yes

Reviewer #2: Yes

Reviewer #1: A few important issues require attention before the paper can be accepted for publication.

1. The rationale for testing only 50% of participants with the Smokerlyzer should be clearly explained. Was this due to budget constraints, time limitations, or another methodological consideration?

2. Several adjusted odds ratios reported in the logistic regression (e.g., age group and sex) have confidence intervals that appear excessively wide or imprecise. Please review these values to confirm their accuracy.

The interpretation of Cohen’s Kappa (κ = 0.048) as “purely coincidental” should be revised to reflect standard interpretations (i.e., "slight agreement").

3. The study reports associations between school environment (lack of health clubs, absence of anti-smoking materials) and smoking behavior. While these are interesting findings, potential residual confounding (e.g., school-level SES, teacher influence, peer smoking norms) should be acknowledged and discussed as a limitation.

Reviewer #2: Reviewing points:

1- The introduction should be revised and rewritten.

2- The introduction does not mention the possible risks and side effects of smoking in adolescents.

3- The study objectives are not well stated at the end of the introduction.

4- The data collection method needs to be revised and explained further.

5- In performing data analysis, in addition to logistic regression, a rank regression analysis should also be considered. Because the response variable of smoking is defined as a rank variable.

6- The entire text of the article should be shortened.

7- Grammatical and reference errors should be corrected.

**Do you want your identity to be public for this peer review?** For information about this choice, including consent withdrawal, please see our For information about this choice, including consent withdrawal, please see our Privacy Policy .

Reviewer #1: **Yes:** Pious AboagyePious Aboagye

Reviewer #2: **Yes:** Solaiman AfroughiSolaiman Afroughi

While revising your submission, please upload your figure files to the Preflight Analysis and Conversion Engine (PACE) digital diagnostic tool, https://pacev2.apexcovantage.com/ . PACE helps ensure that figures meet PLOS requirements. To use PACE, you must first register as a user. Registration is free. Then, login and navigate to the UPLOAD tab, where you will find detailed instructions on how to use the tool. If you encounter any issues or have any questions when using PACE, please email PLOS at . PACE helps ensure that figures meet PLOS requirements. To use PACE, you must first register as a user. Registration is free. Then, login and navigate to the UPLOAD tab, where you will find detailed instructions on how to use the tool. If you encounter any issues or have any questions when using PACE, please email PLOS at figures@plos.org . Please note that Supporting Information files do not need this step.

---

## [Author Response · Author response to Decision Letter 1]

21 Oct 2025

Reviewer 1

S/N COMMENTS REPONSES Page

1 The rationale for testing only 50% of participants with the Smokerlyzer should be clearly explained. Was this due to budget constraints, time limitations, or another methodological consideration?

This was due to financial constraints Page 5

Lines 102-103

2 Several adjusted odds ratios reported in the logistic regression (e.g., age group and sex) have confidence intervals that appear excessively wide or imprecise. Please review these values to confirm their accuracy.

The interpretation of Cohen’s Kappa (κ = 0.048) as “purely coincidental” should be revised to reflect standard interpretations (i.e., "slight agreement").

The values have been reviewed

Page 16-17

Lines 264-265

Page 13-14

Lines 230-232

3 The study reports associations between school environment (lack of health clubs, absence of anti-smoking materials) and smoking behavior. While these are interesting findings, potential residual confounding (e.g., school-level SES, teacher influence, peer smoking norms) should be acknowledged and discussed as a limitation. done Lines 342-345

Pages 21-22

Reviewer 2

S/N COMMENTS REPONSES Page

1

The introduction should be revised and rewritten. Introduction has been revised and rewritten Pages 3-4

Lines 43-76

2 The introduction does not mention the possible risks and side effects of smoking in adolescents.

These have been included

Pages 3-4

Lines 56-65

3. The study objectives are not well stated at the end of the introduction. They have been included Page 4

Line 75-76

4. The data collection method needs to be revised and explained further. This section has been revised and further explained Page 7

Lines 134-151

5. In performing data analysis, in addition to logistic regression, a rank regression analysis should also be considered. Because the response variable of smoking is defined as a rank variable A table on rank regression analysis has been included Page 25

Lines 280-281

6. The entire text of the article should be shortened. Done

7. Grammatical and reference errors should be corrected. Done

8. Additional References have been included while the introduction was rewritten. Lines

365-440

Pages22-25

---

## [Decision Letter · Decision Letter 1]

12 Dec 2025

Dear Dr.  Fehintola,

Thank you for submitting your manuscript to PLOS ONE. After careful consideration, we feel that it has merit but does not fully meet PLOS ONE’s publication criteria as it currently stands. Therefore, we invite you to submit a revised version of the manuscript that addresses the points raised during the review process.

We look forward to receiving your revised manuscript.

Kind regards,

Billy Morara Tsima, MD MSc

Academic Editor

PLOS One

Journal Requirements:

Reviewer's Responses to Questions

**Comments to the Author**

Reviewer #1: (No Response)

Reviewer #2: (No Response)

2. Is the manuscript technically sound, and do the data support the conclusions?

Reviewer #1: Yes

Reviewer #2: Yes

3. Has the statistical analysis been performed appropriately and rigorously?

Reviewer #1: Yes

Reviewer #2: Yes

4. Have the authors made all data underlying the findings in their manuscript fully available?

Reviewer #1: Yes

Reviewer #2: Yes

5. Is the manuscript presented in an intelligible fashion and written in standard English?

Reviewer #1: Yes

Reviewer #2: Yes

Reviewer #1: There are specific technical inconsistencies in the statistical analysis, which were likely introduced during the revision process, and methodological details that require clarification to ensure the validity of the conclusions.

Major Comments

1. Statistical Inconsistency & Multicollinearity (Page 21-22, Tables 7 & 8)

I note that in response to previous reviewer comments, you have included a rank regression analysis (Table 8) alongside the logistic regression (Table 7). However, the results of these two models are contradictory regarding the key predictors: age and class.

In the logistic regression (Table 7), age is a risk factor (AOR=1.59), while class is a protective factor (higher classes have lower odds).

In the Rank Regression (Table 8), age appears protective (negative coefficient), while class is a strong risk factor (positive association).

Action Required: Please check the Variance Inflation Factor (VIF) for age and class in your models. If high multicollinearity exists, you may need to remove one of these variables or combine them to ensure the reported predictors are accurate. You must explain or resolve this contradiction in the results; currently, the conclusion that "older age and class are predictors" is ambiguous because the direction of the effect depends on which table the reader looks at.

Selection Bias in Sub-sample (Methodology Section)

You state that due to financial constraints, 50% of the survey sample was selected for the Smokerlyzer assessment (n=616).

Action Required: Please explicitly clarify the randomization method used to select this 50\% sub-sample. If it was simple random sampling or systematic random sampling within the schools, please state this clearly to ensure the technical soundness of the 27.3% prevalence rate.

Minor Comments

Title Correction

There is a typo in the manuscript title.

Current: "...using Smokerlyze"

Correction: Please change to "...using Smokerlyzer..."

4. Discussion of Age/Class Dynamic

In the Discussion section, you recommend targeted education for "older adolescents" and "lower classes."

Comment: Please briefly elaborate on the interaction between these two demographics. Are the "high risk" students specifically older adolescents who have been held back in lower classes? Clarifying this profile would make your public health recommendations more actionable.

5. Table Formatting

Please ensure the table footnotes for Tables 7 and 8 clearly define the reference categories used for the regression analyses to assist reader interpretation.

Reviewer #2: Reviewing tips for authors and editors

1-In the method section, all subsections from the study site to the exclusion criteria should be combined and written in one subsection.

2- In the sample size determination subsection, the relevant formula is incorrect, correct it.

3- In the final calculation of the sample size, a reference should be mentioned for applying the coefficient of 1.5 based on the category.

4- In the abstract section of the article, there is no mentioned or reported result of the rank regression. The result of the rank regression should be reported or deleted altogether.

5- A reference should be introduced for the use of rank regression.

**Do you want your identity to be public for this peer review?** For information about this choice, including consent withdrawal, please see our For information about this choice, including consent withdrawal, please see our Privacy Policy .

Reviewer #1: No

Reviewer #2: **Yes:** Dr. Solaiman AfroughiDr. Solaiman Afroughi

---

## [Author Response · Author response to Decision Letter 2]

10 Jan 2026

RESPONSE TO THE REVIEWERS COMMENTS

Manuscript Title: Assessment of Carbon Monoxide in Exhaled Breath using Smokerlyze among in-school adolescents in Osun State: A Cross Sectional Study

Reviewer 1

S/N COMMENTS Action required Responses Page

1 Statistical Inconsistency & Multicollinearity (Page 21-22, Tables 7 & 8)

I note that in response to previous reviewer comments, you have included a rank regression analysis (Table 8) alongside the logistic regression (Table 7). However, the results of these two models are contradictory regarding the key predictors: age and class.

In the logistic regression (Table 7), age is a risk factor (AOR=1.59), while class is a protective factor (higher classes have lower odds).

In the Rank Regression (Table 8), age appears protective (negative coefficient), while class is a strong risk factor (positive association).

Please check the Variance Inflation Factor (VIF) for age and class in your models. If high multicollinearity exists, you may need to remove one of these variables or combine them to ensure the reported predictors are accurate. You must explain or resolve this contradiction in the results; currently, the conclusion that "older age and class are predictors" is ambiguous because the direction of the effect depends on which table the reader looks at Thank you for this important observation. We would like to clarify that the apparent difference in direction of association between Tables 7 and 8, arises from differences in outcome definition and model interpretation, rather than statistical inconsistency or multicollinearity.

Table 7 presents a binary logistic regression modelling the likelihood of being a smoker versus a non-smoker, whereas Table 8 applies rank regression to model the relative ordering of smoking behaviour. Rank regression does not estimate risk or severity but rather examines associations with higher or lower ranks of the outcome.

Accordingly, coefficients in Table 8 is not interpreted as indicators of smoking severity or initiation and are not directly comparable with odds ratios from logistic regression.

We have revised the table title, footnote, and manuscript text (Pages 21–22) to explicitly clarify this distinction and to prevent misinterpretation.

In addition, diagnostic checks showed no evidence of problematic multicollinearity among predictors (all VIFs < 2).

Pages 17 -18

Lines 264-275

2 Selection Bias in Sub-sample (Methodology Section)

You state that due to financial constraints, 50% of the survey sample was selected for the Smokerlyzer assessment (n=616). Action Required: Please explicitly clarify the randomization method used to select this 50\% sub-sample. If it was simple random sampling or systematic random sampling within the schools, please state this clearly to ensure the technical soundness of the 27.3% prevalence rate. Done Page 5

Lines 90-97

3 Title Correction

There is a typo in the manuscript title.

Current: "...using Smokerlyze"

Correction: Please change to "...using Smokerlyzer..." Correction: Please change to "...using Smokerlyzer..." Done Line 1

Page 1

4. 4. Discussion of Age/Class Dynamic

In the Discussion section, you recommend targeted education for "older adolescents" and "lower classes."

Comment: Please briefly elaborate on the interaction between these two demographics. Are the "high risk" students specifically older adolescents who have been held back in lower classes? Clarifying this profile would make your public health recommendations more actionable. No,it does not and all VIFs are less than or equal to 2.This does not reveal there are interaction.

It has been corrected

Page 2

Line 38

5. Table Formatting

Please ensure the table footnotes for Tables 7 and 8 clearly define the reference categories used for the regression analyses to assist reader interpretation. The reference categories for Table 7 is indicated as Ref in the table.

Table 8 is a rank regression where all variables are entered as continuous variable and not binary outcome variable, hence no reference categories are presented.

Page 17

Line 261

Reviewer 2

S/N Comments Action required Responses Page

1 In the method section, all subsections from the study site to the exclusion criteria should be combined and written in one subsection. This has been done Page 4-5

Lines 79-88

2 In the sample size determination subsection, the relevant formula is incorrect, correct it. We thank you for this observation. This study was part of a multiphase study where the initial sample size was determined using the formula (Zα + Zβ)² P (1 – P) d² . The estimate of true proportion in the population; was taken as the prevalence of media health literacy among secondary school students in Osun State 37.7%.Hence the formula that was used. A subset of the sample size (50%) was used for the objective measurement of smoking due to the financial constraints

Page 5

Lines 90-97

3. In the final calculation of the sample size, a reference should be mentioned for applying the coefficient of 1.5 based on the category This has been done Page 5

Lines 93-94

4. In the abstract section of the article, there is no mentioned or reported result of the rank regression. The result of the rank regression should be reported or deleted altogether. This has been included Page 2

Line 33-34

5. A reference should be introduced for the use of rank regression. This has been done Page 8

Line 160 and 161

---

## [Editor Report · Decision Letter 2]

7 Apr 2026

Assessment of Carbon Monoxide in Exhaled Breath using Smokerlyzer among in-school adolescents in Osun State: A Cross Sectional Study

PONE-D-25-32127R2

Dear Dr. Fehintola,

We’re pleased to inform you that your manuscript has been judged scientifically suitable for publication and will be formally accepted for publication once it meets all outstanding technical requirements.

Kind regards,

Billy Morara Tsima, MD MSc

Academic Editor

PLOS One
---

## [Editor Report · Acceptance letter]

PONE-D-25-32127R2

PLOS One

Dear Dr. Fehintola,

I'm pleased to inform you that your manuscript has been deemed suitable for publication in PLOS One. Congratulations! Your manuscript is now being handed over to our production team.

Kind regards,

on behalf of

Dr. Billy Morara Tsima

Academic Editor

PLOS One